# Dimerization of Rabies Virus Phosphoprotein and Phosphorylation of Its Nucleoprotein Enhance Their Binding Affinity

**DOI:** 10.3390/v16111735

**Published:** 2024-11-04

**Authors:** Euripedes de Almeida Ribeiro, Cédric Leyrat, Francine C. A. Gérard, Marc Jamin

**Affiliations:** Institut de Biologie Structurale, Université Grenoble Alpes, CEA, CNRS, 38000 Grenoble, France; euripedes.dealmeida@takeda.com (E.d.A.R.J.); cedric.leyrat@igf.cnrs.fr (C.L.); francine.gerard-baraggia@ibcp.fr (F.C.A.G.)

**Keywords:** rabies virus, *Mononegavirales*, nucleoprotein, phosphoprotein, surface plasmon resonance, binding affinity, phosphorylation

## Abstract

The dynamic interplay between a multimeric phosphoprotein (P) and polymeric nucleoprotein (N) in complex with the viral RNA is at the heart of the functioning of the RNA-synthesizing machine of negative-sense RNA viruses of the order *Mononegavirales*. P multimerization and N phosphorylation are often cited as key factors in regulating these interactions, but a detailed understanding of the molecular mechanisms is not yet available. Working with recombinant rabies virus (RABV) N and P proteins and using mainly surface plasmon resonance, we measured the binding interactions of full-length P dimers and of two monomeric fragments of either circular or linear N-RNA complexes, and we analyzed the equilibrium binding isotherms using different models. We found that RABV P binds with nanomolar affinity to both circular and linear N-RNA complexes and that the dimerization of P protein enhances the binding affinity by 15–30-fold as compared to the monomeric fragments, but less than expected for a bivalent ligand, in which the binding domains are connected by a flexible linker. We also showed that the phosphorylation of N at Ser389 creates high-affinity sites on the polymeric N-RNA complex that enhance the binding affinity of P by a factor of about 360.

## 1. Introduction

Rabies is a fearsome zoonotic disease that has been known to humans since prehistoric times [1]. The availability of vaccine and post-exposure treatment together with vaccination campaigns of dogs and wild animals successfully brought the disease under control in Europe and North America [2,3]. Over the years, rabies has become a neglected tropical disease that has an estimated annual economic burden of nearly USD 9 billion and continues to kill more than 50,000 people every year, 40% of whom are young people under the age of 15, mainly in rural areas of Africa and Asia [4,5,6].

The rabies virus (RABV) and other lysaviruses are the etiological agents of this disease. RABV is a neurotropic virus that causes an incurable human brain disease characterized by a complex pathogenesis [7,8,9] and by mechanisms of enzootic maintenance, spread, and evolution that remain poorly understood [10]. RABV is a prototypic member of the *Rhabdoviridae*, a large family of nonsegmented negative-sense RNA viruses (NNVs) (phylum: *Negarnaviricota*—sub-phylum: *Haploviricotina*) that includes the vesicular stomatitis virus (VSV) [11], Chandipura virus [12], and a great number of underexplored viruses infecting various animals and plants [13]. Because they share with many other NNV similar organizations of their genome and virion as well as similar strategies for the transcription, replication, and encapsidation of their genome, the *Rhabdoviridae* are classified in the order *Mononegavirales* with the families *Paramyxoviridae* (including parainfluenza viruses, measles virus, mumps virus, Nipah and Hendra viruses), *Pneumoviridae* (including respiratory syncytial virus and human metapneumovirus), *Filoviridae* (including Ebola and Marburg viruses), and *Bornaviridae* (Borna disease virus) [14,15].

For RABV, as for most NNV members, the infectious ribonucleoprotein complex is composed of the viral RNA genome and three viral proteins: the nucleoprotein (N), the phosphoprotein (P), and the large RNA-dependent RNA polymerase (L) (Figure 1A). More than 1300 N molecules, each one coating nine nucleotides, assemble into a linear homo-polymer that encapsidates the genomic RNA, forming a ribonucleoprotein complex named as a nucleocapsid (NC) [16]. The L polymerase uses the NC as a template for both transcription and replication and carries out RNA synthesis as well as 3′ and 5′ processing of messenger RNAs [17]. P is a multifunctional protein with a modular architecture [18], which with its different naturally truncated forms [19] plays essential roles in viral RNA transcription and replication but also in hijacking cellular machineries [20] and escaping innate immune responses [21,22,23]. P forms dimers [24,25], and each protomer (297 aa) consists of a long N-terminal intrinsically disordered region (aa 1–88) and a C-terminal region made of two folded domains, the dimerization domain, MD (aa 89–132), and the C-terminal domain (aa 194–295), CTD, which are connected by a flexible linker (aa 133–193) [18,20,25,26] (Figure 1B,C). As part of its role in viral replication, P works as a hub, in which structural and functional modules interact with their different partners (Figure 1C): (i) by a module (aa 1–39) located at its N-terminus, and P acts as a chaperone of unassembled RNA-free nucleoprotein (N^0^) [27,28,29,30,31] (Figure 1D); (ii) by a module encompassing the last part of the N-terminal region (aa 51–88), and P binds to L [32], and by its CTD, it binds to the NC [26,33,34], tethering the polymerase to its template and ensuring its processivity during RNA synthesis [17] (Figure 1E,F); (iii) by being dimeric and forming weak multivalent interactions with itself and with N, and P acts as a scaffold protein for the formation of membrane-less viral factory compartments (MLO) by liquid–liquid phase separation [35,36]. In our previous study, we generated a model of P CTD bound to a circular polymeric N-RNA complex, where the binding site for P CTD involved two adjacent N protomers (Figure 1F) [33]. P CTD lies on the top of the N_CTD_ of the N_i_ protomer, while by an induced-fit mechanism, the N_CT-ARM_ of the same protomer (N_i_) and that of the adjacent one (N_i__−1_) mold around the P CTD, making extensive protein–protein contacts.

The dynamic interplay between P, N, and L is at the heart of the functioning of the RNA polymerase machine. In NC, the N proteins sequester the genomic RNA, suggesting that, during both the transcription and replication of the genome, a short RNA segment must transiently extract from the N homo-polymer, forming a bubble that allows the polymerase access to the bases [16] (Figure 1A). Also, the L polymerase and its cofactor P must move along the RNA template, displacing the transcription/replication bubble, while during replication, the chaperone P must deliver its N^0^ cargo to the site of RNA synthesis and transfer N^0^ to the nascent RNA molecule (Figure 1A). Understanding the mechanisms of this machine thus requires a quantitative characterization of the stability and dynamics of these different molecular interactions and the identification of the structural features that control them. In particular, two features are regularly put forward as key regulators of these interactions, namely the mutimerization of P and the phosphorylation of P or N.

In all NNVs, the P protein is multimeric, forming dimers in *Rhabdoviridae* [24,25,37] and tetramers in all other families [38,39,40,41]. It has been proposed that multimerization is essential for allowing P, in association with L, to cartwheel along polymeric N during RNA synthesis [42]. In such a model, each P_CTD_ would dissociate from the template one at a time and, following the displacement of the polymerase complex, re-associate downstream of their initial position. The alternate binding and release of the different P_CTD_ would allow the polymerase complex to almost permanently remain attached to the template NC [43].

Similarly, the phosphorylation of P and/or N is considered as an important regulation mechanism of the RNA polymerase complex in different NNVs [44,45,46], although the establishment of a molecular mechanism clearly explaining the role of phosphorylation is still lacking for any of them. In the case of RABV, N is the most phosphorylated protein among the proteins present in the virion [44] or when expressed in insect cells as a recombinant protein [47]. The major phosphorylation site is Ser389, which is localized within a flexible loop of the C-terminal domain of N, named N_CT-ARM_ [48]. This phosphorylation, catalyzed by casein kinase II [49], is essential for both viral transcription and replication [45,46] and for regulating the encapsidation process and the assembly of NCs [45,50]. It has been demonstrated that the phosphorylation of Ser389 in N is not required for the formation of the N^0^-P complex [50,51] and that the phosphorylation of N occurs only after the formation of NC [50], enhancing the stability of the complex between NC and the C-terminal domain of P [51].

Here, we set out to test how the dimerization of P and the phosphorylation of N affect the affinity of RABV P for the NC. To these ends, we used equilibrium surface plasmon resonance (SPR) experiments and measured binding affinities between recombinant full-length dimeric RABV P (P_2_) and circular or linear N-RNA complexes. We found that the P dimer binds with an affinity that is 15–30-fold larger than that of the monomers, but with no evidence of strong cooperativity. We also showed that the phosphorylation of N enhances the affinity for P by a factor of about 360. We discuss the importance of these results for the operation of the RABV RNA synthesis machine.

## 2. Materials and Methods

### 2.1. Sample Preparation

Recombinant N-RNA rings, full-length P, P_Δ90–133_, and the C-terminal domain of P (P_CTD_) were produced and purified as previously described [34,35,36]. RABV nucleocapsids were produced and purified from infected BSR cells as described previously [52]. P protein concentrations were measured by absorbance spectroscopy using Edelhoch’s method [53]. N-RNA concentrations were measured by integrating the elution peak of the SEC profile monitored by refractometry and using a dn/dc value of 0.185 mL·g^−^^1^. Each sample used was tested by SEC-MALLS-RI for the homogeneity of the preparation.

### 2.2. Native Gel Electrophoresis

A native gel of 4% acrylamide with a 19:1 acrylamide/bisacrylamide ratio was pre-run for 30 min at 200 V in a cold room. Typically, 10 µL of the protein mixture was loaded, and the gel was run for 5 h at 200 V and 4 °C in Tris-Borate-EDTA (0.2×), pH 8.0. Proteins were identified by Coomassie blue staining.

### 2.3. Western Blotting

The protein samples were separated by SDS polyacrylamide gel electrophoresis (SDS-PAGE), using a 10% acrylamide slab gel with a 5% acrylamide stacking gel, in a Mini-Protean II Dual Slab Cell (Bio-Rad, Hercules, CA, USA). The current was set at 25 mA, and the gels were run for 2 h. After electrophoresis, the gels were blotted onto 0.45 µm nitrocellulose membranes (Schleicher and Schuell, Keene, NH, USA) for 1 h, at 20 V, using a Trans-Blot semi-dry transfer cell. The membranes were then blocked by incubation for 45 min with 5% BSA in Tris-buffered saline (TBS: 20 mM Tris-HCl, pH 7.6, 150 mM NaCl) containing 0.1% Tween 20 (TBS-T), and then washed twice with TBS-Tween. The protein blots were probed with an IgG Rabbit polyclonal anti-phosphoserine antibody conjugated to biotin at 0.5 µg·mL^−1^ (abcam^®^, ab9335, Cambridge, UK) in TBS-T 5% BSA. After overnight incubation at 4 °C, the blots were washed three times with TBS-T for 10 min at room temperature followed by treatment with a streptavidin–horseradish peroxidase conjugate (Invitrogen, Waltham, MA, USA) at 0.05 µg·mL^−^^1^ in TBS-T 5% BSA for 1 h at room temperature. The blots were then washed four times with TBS-T for 15 min at room temperature and the band detection was made using a chemiluminescence procedure (SuperSignal West Pico Chemiluminescent substrate, Thermo Scientific, Waltham, MA, USA). Blots were exposed to an imaging plate for 2 min.

### 2.4. Surface Plasmon Resonance

Analyses were carried out on a Biacore X (GE Healthcare, Chicago, IL, USA). The RABV N_10_-RNA complex was immobilized on a CM5 sensor chip using an amine coupling kit (GE Healthcare) to final resonance values of about 4000 response units (RU). RABV P or P fragments were in a 20 mM Tris-HCl buffer at pH 7.5 containing 150 mM NaCl, 1 mM DTT, and 0.005% (*v*/*v*) Tween20. Data were collected at 20 °C at a flow rate of 20 µL·min^−^^1^. Binding curves were corrected for background and bulk refractive index contribution by subtracting the reference flow cell. The regeneration of the surface was achieved by the injection of 20 µL of a 20 mM Tris-HCl buffer at pH 7.5 containing 1 M NaCl at a flow rate of 50 µL·min^−1^. Over the course of the experiment, the amount of immobilized N_10_-RNA decreased by 10–20% between the first and last P_CTD_ injections. The binding at each ligand (δ_mes_) concentration was adjusted (δ_corr_) for the level of N_10_-RNA on the surface immediately preceding that injection (RU_N10-RNA_)_initial_ using the equation
(1)$$\delta_{corr}=\delta_{mes}\frac{{\left({RU}_{N10-RNA}\right)}_{initial}}{{\left({RU}_{N10-RNA}\right)}_{injection}}$$
where (RU_N10-RNA_)_initial_ is the level of immobilized N_10_-RNA immediately preceding the first injection of P_2_. The equilibrium resonance measurements were fitted to a simple Langmuir binding isotherm,
(2)$$\delta_{corr}=\frac{{\left(RU\right)}_{max}\left[L\right]}{K_{d,app}+\left[L\right]}$$
where $\delta_{corr}$ is the adjusted signal measured in RUs, (RU)_max_ is the maximum response, $\left[L\right]$ is the concentration of the ligand, and K_d,app_ is the apparent dissociation constant, or, two, a two-term Langmuir binding isotherm,
(3)$$\delta_{corr}=\frac{{\left(RU\right)}_{max1}\left[L\right]}{K_{d,app1}+\left[L\right]}+\frac{{\left(RU\right)}_{max2}\left[L\right]}{K_{d,app2}+\left[L\right]}$$
where (RU)_max1_ and (RU)_max2_ are the maximum responses and K_d,app1_ and K_d,app2_ are the apparent dissociation constants. The stoichiometry of the complexes was calculated from the ${\left({RU}_{N10-RNA}\right)}_{initial}$ and (RU)_max_ values using
(4)$$n=\frac{{\left(RU\right)}_{max}}{{\left({RU}_{N10-RNA}\right)}_{initial}}.\frac{{MM}_{N10-RNA}}{{MM}_{L}}$$
where ${MM}_{N10-RNA}$ is the molecular mass of the N_10_-RNA complex (530 kDa) and ${MM}_{L}$ is the molecular mass of the ligand, either P_CTD_, P_Δ90–133_, or P_2_.

### 2.5. Modeling Equilibrium Binding Isotherms

#### 2.5.1. Binding of Monomeric Ligand L to Independent Sites (Binding Model 1)

If we consider a system in which two sites (denoted by the subscripts a and b) of the same homo-polymeric N-RNA complex (named M) bind independently of each other, the following applies to monomeric ligand L:$${\;\;K}_{a}\;M+L\;⇋\;{\mathrm{ML}}_{a}\;\;\;{\;\;K}_{b}\;M+L\;⇋\;{\mathrm{ML}}_{b}$$

The dissociation constants for each site are given by
$$K_{a}=\frac{\left[M\right]\left[L\right]}{\left[ML_{a}\right]}K_{b}=\frac{\left[M\right]\left[L\right]}{\left[ML_{b}\right]}$$

The total degree of binding $\bar{\nu }$ (i.e., moles of L bound per moles of M) is given by the sum of the amounts bound to each site:(5)$$\bar{\nu }=\frac{\frac{\left[L\right]}{K_{a}}}{1+\frac{\left[L\right]}{K_{a}}}+\frac{\frac{\left[L\right]}{K_{b}}}{1+\frac{\left[L\right]}{K_{b}}}$$

If each site binds L with the same affinity, the dissociation constants are the same: K_a_ = K_b_ = K 
and
(6)$$\bar{\nu }=\frac{2\frac{\left[L\right]}{K}}{1+\frac{\left[L\right]}{K}}$$

The fraction of sites filled by L is given by $\bar{\nu }$ and the fraction of empty sites is given by $2-\bar{\nu }$. The ratio of filled to empty sites is then given by
(7)$$\frac{\bar{\nu }}{2-\bar{\nu }}=\frac{2\frac{\left[L\right]}{K}}{2+2\frac{\left[L\right]}{K}-2\frac{\left[L\right]}{K}}=\frac{\left[L\right]}{K}\;$$

The Hill plot is obtained by plotting $log\left(\frac{\bar{\nu }}{2-\bar{\nu }}\right)$ against $\log\left[L\right]$.

#### 2.5.2. Binding of One Dimeric Ligand L to a Homo-Polymeric Complex (Binding Model 2)

If we consider a system in which two sites of the same homo-polymeric N-RNA complex bind independently, the two binding domains of a dimeric ligand L, the first binding reaction is bimolecular, whereas the second is intramolecular (denoted by superscript i):$${\;\;\;K}_{a}\;\;\;\;\;{K_{b}}^{i}\;M+L\;⇋\;{\mathrm{ML}}_{a}\;⇋\;{\mathrm{ML}}_{\mathrm{ab}}$$

The dissociation constants are given by
$$K_{a}=\frac{\left[M\right]\left[L\right]}{\left[ML_{a}\right]}K_{b}^{i}=\frac{\left[ML_{a}\right]}{\left[ML_{b}\right]}$$

The total degree of binding $\bar{\nu }$ is given by
(8)$$\bar{\nu }=\frac{\left[ML_{a}\right]+\left[ML_{b}\right]\;}{\left[M\right]+\left[ML_{a}\right]+\left[ML_{b}\right]}=\frac{\frac{\left[L\right]}{K_{a}}+\frac{\left[L\right]}{K_{a}K_{b}^{i}}}{1+\frac{\left[L\right]}{K_{a}}+\frac{\left[L\right]}{K_{a}K_{b}^{i}}}$$

The ratio of filled to empty sites is then given by
(9)$$\frac{\bar{\nu }}{1-\bar{\nu }}=\frac{\frac{\left[L\right]}{K_{a}}+\frac{\left[L\right]}{K_{a}K_{b}^{i}}}{1+\frac{\left[L\right]}{K_{a}}+\frac{\left[L\right]}{K_{a}K_{b}^{i}}-\frac{\left[L\right]}{K_{a}}-\frac{\left[L\right]}{K_{a}K_{b}^{i}}}=\frac{\left[L\right]}{K_{a}}+\frac{\left[L\right]}{K_{a}K_{b}^{i}}=\frac{\left(1+K_{b}^{i}\right)}{K_{a}K_{b}^{i}}\left[L\right]$$

#### 2.5.3. Binding of Two or More of Dimeric Ligand L to a Homo-Polymeric Complex (Binding Model 3)

If we consider a system in which two dimeric ligands bind independently to a homo-polymeric N-RNA complex,
$${\;\;\;K}_{a1}\;\;\;\;\;{K_{b1}}^{i}\;M+L\;⇋\;{\mathrm{ML}}_{a1}\;⇋\;{\mathrm{ML}}_{\mathrm{ab}1}\;\;\;{\;\;\;K}_{a2}\;\;\;\;\;{K_{b2}}^{i}\;M+L\;⇋\;{\mathrm{ML}}_{a2}\;⇋\;{\mathrm{ML}}_{\mathrm{ab}2}$$

The dissociation constants are given by
$$K_{a1}=\frac{\left[M\right]\left[L\right]}{\left[ML_{a1}\right]}K_{b1}^{i}=\frac{\left[ML_{a1}\right]}{\left[ML_{b1}\right]}$$
$$K_{a2}=\frac{\left[M\right]\left[L\right]}{\left[ML_{a2}\right]}K_{b2}^{i}=\frac{\left[ML_{a1}\right]}{\left[ML_{b2}\right]}$$

The total degree of binding $\bar{\nu }$ is given by the sum of the amounts of each ligand bound:(10)$$\bar{\nu }=\frac{\frac{\left[L\right]}{K_{a1}}+\frac{\left[L\right]}{K_{a1}K_{b1}^{i}}}{1+\frac{\left[L\right]}{K_{a1}}+\frac{\left[L\right]}{K_{a1}K_{b1}^{i}}}+\frac{\frac{\left[L\right]}{K_{a2}}+\frac{\left[L\right]}{K_{a2}K_{b2}^{i}}}{1+\frac{\left[L\right]}{K_{a2}}+\frac{\left[L\right]}{K_{a2}K_{b2}^{i}}}$$

The ratio of filled to empty sites is then given by
(11)$$\frac{\bar{\nu }}{2-\bar{\nu }}=\frac{\frac{\left[L\right]}{K_{a1}}+\frac{\left[L\right]}{K_{a2}}+\frac{\left[L\right]}{K_{a1}K_{b1}^{i}}+\frac{\left[L\right]}{K_{a2}K_{b2}^{i}}+\frac{2\;{\left[L\right]}^{2}}{{K_{a1}K}_{a2}}+\frac{2\;{\left[L\right]}^{2}}{K_{a1}{K_{a2}K}_{b1}^{i}}+\frac{2\;{\left[L\right]}^{2}}{{K_{a1}K}_{a2}K_{b2}^{i}}+\frac{2{\;\left[L\right]}^{2}}{K_{a1}K_{a2}K_{b1}^{i}K_{b2}^{i}}}{2+\frac{\left[L\right]}{K_{a1}}+\frac{\left[L\right]}{K_{a2}}+\frac{\left[L\right]}{K_{a1}K_{b1}^{i}}+\frac{\left[L\right]}{K_{a2}K_{b2}^{i}}}$$

Equations (6) and (10) can be combined to any number of independent binding sites. To reproduce the isotherm for P_2_ binding to N_10_-RNA, we considered the binding of two P_2_ via two CTD to high-affinity phosphorylated sites with different dissociation constants and two P_2_ via only one CTD to low-affinity non-phosphorylated sites (using a dissociation constant $K_{a3}$:(12)$$\bar{\nu }=\frac{\frac{\left[L\right]}{K_{a1}}+\frac{\left[L\right]}{K_{a1}K_{b1}^{i}}}{1+\frac{\left[L\right]}{K_{a1}}+\frac{\left[L\right]}{K_{a1}K_{b1}^{i}}}+\frac{\frac{\left[L\right]}{K_{a2}}+\frac{\left[L\right]}{K_{a2}K_{b2}^{i}}}{1+\frac{\left[L\right]}{K_{a2}}+\frac{\left[L\right]}{K_{a2}K_{b2}^{i}}}+2\left(\frac{\frac{\left[L\right]}{K_{a3}}}{1+\frac{\left[L\right]}{K_{a3}}}\right)$$

#### 2.5.4. Calculation of SPR Signal

In each case, if the binding of one ligand L generates an SPR signal ${\Delta RU}_{L}$, the SPR signal for the entire binding isotherm is given by
(13)$$\delta_{RU}={\Delta RU}_{L}\cdot \bar{\nu }$$

## 3. Results

### 3.1. Binding of P Dimers to Phosphorylated Circular N_10_-RNA Complex

We expressed recombinant RABV N in insect cells yielding linear N-RNA complexes resembling viral nucleocapsids and different circular N-RNA complexes containing varying numbers of N protomers [54] (Figure 2A, lane 1). We purified the circular complex containing 10 N protomers (N_10_-RNA) by preparative electrophoresis (Figure 2A, lane 3) and confirmed its identity by negative staining electron microscopy as previously described [16,54]. We showed by SDS-PAGE followed by a Western blot using anti-phosphoserine antibodies that the N protein in the recombinant N_10_-RNA complex was phosphorylated in agreement with previous reports [47,50] (Figure 2B,C). By incubating the complex with calf intestinal alkaline phosphatase (CIAP) for 2 h, we almost completely dephosphorylated the protein as shown by immunolabeling (Figure 2C), while we confirmed the presence of similar amounts of N in both samples by Coomassie staining (Figure 2B). We could subsequently re-phosphorylate the N protein with casein kinase II (CKII).

To obtain a semi-quantitative estimate of the level of the phosphorylation of N in the N_10_-RNA complex, we used peptide mass spectrometry fingerprinting. We analyzed samples of N_10_-RNA treated or non-treated with thermo-sensitive alkaline phosphatase (FastAP) by SDS PAGE with staining by Coomassie blue. Subsequently, the protein was treated with trypsin by in-gel digestion, and the peptides were analyzed by electrospray Q-TOF mass spectrometry with 95% sequence coverage. In both samples, we found the peptide corresponding to residues 377 to 400 of N (SDVALADDGTVNSDDEDYFSGETR) and containing Ser389 in both its unphosphorylated (*m*/*z* 860.0 [M+H]^2+^; *m*/*z* 1289.5 [M+H]^3+^) and phosphorylated forms (*m*/*z* 886.6 [M+H]^2+^; *m*/*z* 1329.5 [M+H]^3+^) (Figure 2D). In the non-treated sample, peak area ratios of 2.7 and 3.1 were found between the phosphorylated and unphosphorylated peptides for the doubly and triply charged ions, respectively. No other peptide containing a phosphorylated residue was detected in agreement with previous data [48]. After treatment with FastAP, the same unphosphorylated and phosphorylated forms were found, with intensity ratios 0.005 and 0.030 for the doubly and triply charged ions, respectively (Figure 2D).

These results demonstrated that about 7 out of 10 N molecules on average were phosphorylated on Ser389 in the N_10_-RNA complex produced in insect cells. It also showed that only a low fraction of the N molecules (less than 5%) remained phosphorylated after phosphatase treatment.

Full-length RABV P was produced and purified as previously described [24] (Figure 2A, lane 2). The protein forms dimers (P_2_) [24,25], and our previous results showed (i) that the complexes formed between P_2_ and circular N_10_-RNA sustained separation by size exclusion chromatography (SEC), suggesting a submicromolar affinity, and (ii) that at least two P_2_ bind to the N_10_-RNA complex in successive steps [33]. Here, we monitored binding of full-length dimers to the circular N_10_-RNA complex by native gel electrophoresis (Figure 2A). The N_10_-RNA complex appeared as a single band in the gel. At sub-stoichiometric concentrations of P_2_ (using a N_10_-RNA concentration of 2 μM), a first complex (complex I) was detected as a distinct up-shifted band. At increasing [P_2_]/[N_10_-RNA] ratios, the amount of free N_10_-RNA decreased whereas the amount of complex I increased. A second complex was detected as an upper-shifted band (complex II) in agreement with the binding of a second dimer of P [33]. No excess of P_2_ was detected, indicating that the concentrations used in these experiments were above the dissociation constants of the complexes. At [P_2_]/[N_10_-RNA] ratios up to 1.0, only the bands corresponding to complexes I and II and to N_10_-RNA alone were visible, but at higher ratios, a third complex of even larger size and lower affinity (complex III) appeared, which had not been detected previously by SEC [33]. SEC experiments result in large sample dilution, and because of the low stability of complex III, it is likely that it dissociated during elution. These results confirmed that complexes with one (complex I) or two P dimers (complex II) bound per N_10_-RNA ring form in a concentration-dependent process and that these two complexes are the major species present at [P_2_] ≤ 2 µM.

### 3.2. Phosphorylation of N Creates High-Affinity Binding Sites for P

The binding of P_2_ to the recombinant circular N_10_-RNA complex was then monitored by surface plasmon resonance (SPR). The N_10_-RNA complex was immobilized on a CM5 sensor chip, while P_2_ was injected through the flow cell. The sensorgrams (Figure 3A) revealed (i) that the association of P_2_ with the N_10_-RNA complex was slow and that equilibrium was not reached after 200 s at P_2_ concentrations lower than 100 nM, and (ii) that the dissociation behavior, when the buffer flowed through the chip, depended on P_2_ concentration in the association step. At low P_2_ concentrations, almost no dissociation process was measurable on a typical SPR experiment timescale, whereas at increasing P_2_ concentrations, one part of the complex dissociated with measurable kinetics, while the remaining part still dissociated too slowly to be monitored (Figure 3A). The proportion of the complex that dissociated with measurable kinetics increased with P_2_ concentration. Thus, at 500 nM P_2_, the observable dissociation process represented about 60% of the total signal and followed bi-exponential kinetics with off-rates of 0.122 ± 0.006 s^−1^ (A_1_ = 30%) and of 0.015 ± 0.001 s^−1^ (A_2_ = 30%), suggesting a multi-step process or the presence of different species in the sample (Figure 3B). For 40% of the apparently non-dissociating complex, an upper limit of 0.0001 s^−1^ was estimated for the off-rate constant. Similar kinetics were obtained when P_2_ was immobilized on the sensor chip, while the N_10_-RNA complex was injected through the flow cell.

After treatment of the N_10_-RNA complex with alkaline phosphatase in situ on the sensor chip for 2 h at 20 °C, the association process remained similar but 95% of the complex population dissociated with measurable bi-exponential kinetics (k_off_ = 0.25 ± 0.01 s^−1^ (A_1_ = 71%) and 0.021 ± 0.001 s^−1^ (A_2_ = 25%)) (Figure 3C). Only about 5% of the population of bound P_2_ still dissociated too slowly to be seen. The reduction in the proportion of the slowly dissociating complex clearly indicated that the dephosphorylation of N eliminated high-affinity binding sites for P_2_. A subsequent in situ treatment of the dephosphorylated N_10_-RNA complex with CKII restored the dissociation kinetics observed with the initial phosphorylated material, nearly the same proportions of bound P_2_ dissociating in the measurable kinetic phases (66%) and dissociating too slowly to be measured (34%) (Figure 2D) compared to in the initial sample (Figure 3B).

The results of these SPR experiments, in agreement with the native gel experiment, supported the existence of different populations of bound P_2_ molecules. At all P_2_ concentrations, both the association and dissociation processes were multiphasic and could not be fitted with a simple binding isotherm, and no attempt was made to derive a complex mathematical model that could reproduce these kinetic data. P_2_ molecules that dissociated with measurable kinetics corresponded to molecules bound with lower affinities than those that dissociated with kinetics too slow to be measured. These populations increased from 60% to 100% upon the dephosphorylation of N, suggesting that these kinetic phases corresponded to the dissociation of P_2_ from non-phosphorylated sites of N_10_-RNA and corresponded to complex III on native gel. Given their fast dissociation, it is likely that a large part of complex III dissociated during electrophoresis, possibly explaining the presence of complex III only at the larger [P_2_]/[N_10_-RNA] ratios and the existence of the smear between the three complexes seen on the gel (Figure 2A). The population of P_2_ molecules that dissociated with a very slow off-rate indicated binding with high affinity. The size of this population increased upon the phosphorylation of N and must include complexes I and II.

### 3.3. Equilibrium SPR Analysis

Because both association and dissociation processes exhibited complex kinetic behaviors that could not be fitted with simple binding models, and because both association and dissociation kinetics were slow, equilibrium measurements were obtained by performing multiple sequential injections of the ligand until equilibrium was reached. Figure 4 shows a typical sensorgram obtained for a series of sequential injections. In this example, the response signal reached equilibrium after six injections of the P_2_ solution. The response at equilibrium was measured at increasing P_2_ concentrations, and these values were used to draw binding isotherms and determine the stoichiometry and binding affinity.

### 3.4. Dimerization Increases the Affinity of P for the N-RNA Template

To test the effect of P dimerization on the binding affinity for the N_10_-RNA complexes, we established equilibrium binding isotherms for P_2_ and for the truncated P_Δ90–133_ and compared them with the isotherm previously determined under identical conditions for the C-terminal domain of P (P_CTD_) [33] (Figure 1C and Figure 5a).

For both P fragments (P_CTD_ and P_Δ90–133_), which were shown to be monomeric [18,33], the binding isotherm reached saturation at concentrations above 1 µM (1000 nM). For full-length P_2_, a first plateau was reached near 1 µM, but additional binding events occurred at higher concentrations (Figure 5a). Saturation for this second process could not be reached in our experiments because P_2_ could not be prepared at higher concentrations. This additional binding process occurred in the same concentration range and thus likely corresponded to the formation of complex III detected by native gel electrophoresis (Figure 2A). Similar data for the interaction between P_2_ and N_10_-RNA were obtained in the presence of 500 mM NaCl (Figure 5a), ruling out non-specific binding to the chip.

Assuming that in the SPR experiments all bound N_10_-RNA complexes were accessible for ligand binding, the stoichiometry of the complexes was readily calculated from the amplitudes of the response change at saturation. In these experiments, the level of N_10_-RNA immobilized was 4600 ± 50 R.U. At saturation, binding of P_CTD_ and of P_Δ90–133_ reached levels of 260 ± 10 R.U. and 430 ± 30 R.U. above the initial signal, respectively, indicating stoichiometries of 2.0 ± 0.2 and 1.7 ± 0.2 monomers of the P fragment bound per N_10_-RNA ring, respectively. At a P_2_ concentration of 1 μM, the signal reached a level of 1040 ± 30 R.U., indicating a stoichiometry of 3.5 ± 0.2 monomers of P bound per N_10_-RNA ring, close to 4 being expected for the binding of two dimers. Thus, monomeric RABV P_CTD_ and P_Δ90–133_ bound to the N_10_-RNA ring with a stoichiometry of two monomers per ring, whereas full-length P_2_ bound with a stoichiometry of two dimers per ring (four P monomers per ring) at a concentration of 1 μM in agreement with the values obtained previously by SEC titration at analyte concentrations above saturation [33].

In our experimental conditions, the binding of individual P_CTD_, P_Δ90–133_, or P_2_ molecules was not resolved and, within experimental errors, the binding of two molecules could be reproduced up to 1 μM with a single Langmuir binding isotherm (Equation (2)), yielding an apparent dissociation constant, K_d,app_, of 160 ± 20 nM for P_CTD_, 80 ± 10 nM for P_Δ90–133_, and 5.5 ± 1.0 nM for full-length P_2_ (Figure 5a). Figure 5b shows the normalized isotherms, and highlights that binding of P_2_ to the N_10_-RNA complex was 15- to 30-fold stronger than binding of monomeric fragments. We also note that the affinity of P_Δ90–133_ is twice that of P_CTD_, suggesting that somehow the long N-terminal disordered region contributes to P binding to the NC. The Hill plots calculated from the fitted curves (Figure 5c) yielded Hill coefficients of 1.09 ± 0.06, 0.92 ± 0.05, and 0.86 ± 0.05 for P_CTD_, P_Δ90–133_, and P_2_, respectively, indicating poor (for P_2_) or no binding cooperativity (for P_CTD_ and P_Δ90–133_).

Accordingly, the binding isotherms for both P fragments could also be reproduced by assuming that two monomers bind independently of each other, each with the same single-site binding constant K (binding model 1). In line with this model, the amount of ligand bound is the sum of the amount of ligand bound to each site (Equation (7)), and the degree of binding (Hill coefficient) is given by $\frac{\left[L\right]}{K}\;$(Equation (8)) in good agreement with the data (Figure 5c).

To reproduce the binding isotherm for P_2_ and explain the native gel results (Figure 2A), we assumed that two P_2_ bound with high affinity (binding below 1 μM) to a single N_10_-RNA complex via their two CTDs (the name CTD refers to the C-terminal domain (aa 194–295), whether isolated or part of P_Δ90–133_ or full-length P_2_, whereas the name P_CTD_ refers to the fragment of P containing only the C-terminal domain) and that additional P_2_ bound with lower affinity (binding above 1 μM). Within P_2_, the two CTD are connected by flexible linkers to the dimerization domain, and thereby are connected to each other. Thus, once the first CTD is bound to one N_10_-RNA complex, the binding of the second CTD is intramolecular instead of bimolecular, providing an entropic advantage [55]. Because the phosphorylation of N_10_-RNA significantly enhanced the binding affinity and that 7 out of 10 N were phosphorylated on average in the circular complex, we assumed that the first two P_2_ bound to phosphorylated sites. If we assumed that the first two P_2_ bound completely independently of each other (binding model 3), i.e., each with the same $K_{a}$ (80 μM) and $K_{b}^{i}$ (0.03), the predicted Hill coefficient (Equation (11)) slightly diverged from the data (Figure 5c) possibly because the second CTD had less binding sites available. A better fit could be obtained by using the model (binding model 3) in which the first P_2_ bound with $K_{a1}$ = 80 μM and $K_{b1}^{i}$= 0.03 (which yielded a macroscopic dissociation constant of $K_{a1}\cdot K_{b1}^{i}$ = 2.4 μM), while the second P_2_ bound with $K_{a2}\;$= 560 μM and $K_{b2}^{i}$= 0.03 (which yielded a macroscopic dissociation constant of $K_{a2}\cdot K_{b2}^{i}$ = 16.8 μM). This model reproduced the experimental binding isotherm (Equation (10)) and Hill plot (Equation (11)) up to 1 μM slightly better than the independent-site model (Figure 5a,c) and also provided an explanation for the appearance of complex I (one bound P_2_) at lower P_2_ concentrations compared to complex II (two bound P_2_) in the native gel (Figure 2A).

To explain the additional binding observed above 1 μM (Figure 5a), the two kinetic observable phases in the SPR experiments (Figure 3B), and the presence of complex III in the native gel (Figure 2A), we considered that one or two additional molecules of P_2_ were binding to the ring with lower affinities. Because two P_2_ molecules were already bound with high affinity to phosphorylated sites, and the apparent dissociation constant was not very different from that determined for P_2_ binding to the dephosphorylated N_10_-RNA complex (see below), we hypothesized that one or two additional P_2_ molecules bound independently to non-phosphorylated sites. However, it is also possible that they bound by another mechanism involving, for example, the N-terminus of P. By adding a third term to our binding equation (Equation (12)), we could reproduce the entire binding isotherm (the dashed line in Figure 5a was calculated assuming the binding of two additional P_2_ molecules).

### 3.5. Interactions with Linear NCs

Similarly, we studied the binding of full-length P_2_ or of the truncated forms of P to linear RABV NCs purified from insect cells. In contrast to the circular N_10_-RNA complex, samples of linear NC were heterogeneous in size, and therefore the amplitude of the response signal could not be used to estimate the stoichiometry of the complexes. We also did not determine the degree of the phosphorylation of N in these samples. As with the N_10_-RNA circular complexes, the association kinetics were slow at low P_2_ concentrations and the rate increased with increasing P_2_ concentration. At 1 µM P_2_, the equilibrium was reached in less than 200 s, and the dissociation of the complex occurred in two different phases. About 50% of the complex disappeared in a measurable reaction, while 50% dissociated too slowly to be monitored. The presence of two kinetic processes of similar amplitude occurring on two different timescales also indicated the presence on the linear NCs of binding sites of different affinities for P_2_. No evidence of additional binding, however, was found at P_2_ concentrations higher than 1 µM as found with the N_10_-RNA complex.

The binding isotherms for full-length P_2_ but also for P_CTD_ and for P_∆90-133_ spread in a broader concentration range than those obtained with the N_10_-RNA rings (Figure 6a). For all protein, but mainly for P_2_ and P_CTD_, the binding isotherm clearly exhibited two successive and distinct saturation processes within the accessible concentration range, and they could be reproduced by an equation containing two binding terms (Equation (3)). The interactions with linear NCs thus appeared to follow similar binding schemes compared to the N_10_-RNA rings, although without knowing neither the number of ligand molecules that bind in each binding step or the stoichiometry of the different bound forms, it was difficult to further analyze the mechanism. However, the comparison of the normalized binding curves clearly revealed that about one half of dimeric full-length P molecules bound with a higher affinity than the other half, and compared to the monomeric fragments (Figure 6b). The macroscopic dissociation constants K_d,app1_ = 0.9 ± 0.3 nM and K_d,app2_ = 135 ± 60 nM for full-length P_2_ determined by non-linear regression for the high- and low-affinity sites indicated that P_2_ bound even stronger to the high-affinity sites of NCs than to those of the circular N_10_-RNA complex. This may reflect geometrical constraints and/or steric hindrance in the circular complex.

### 3.6. Phosphorylation Strongly Enhances the Binding Affinity of P Dimers

To quantify the effect of N phosphorylation on the binding affinity for full-length P_2_, we also measured the binding curves for the dephosphorylated N_10_-RNA complex and circular NC obtained by in situ treatment with CIAP. Loss of material during the treatment with CIAP prevented the direct comparison of the amplitudes, which are therefore reported in their normalized form.

The binding curve obtained with the dephosphorylated N_10_-RNA complex was rather well reproduced by considering that 5% of N molecules remained phosphorylated and bound P_2_ with a high affinity (using the same values for the parameters $K_{a1}$, $K_{b1}^{i}$, $K_{a2}$, and $K_{b2}^{i}$ as for the fit with phosphorylated N_10_-RNA complexes) and fitting the remaining part with a single binding isotherm and an apparent value K_d,app_ of 2000 ± 300 nM. The binding affinity of P_2_ to the dephosphorylated N_10_-RNA complex is thus 360-fold weaker than that of P_2_ for the phosphorylated complex (Figure 7a). A similar value of K_d,app_ of 1600 ± 300 nM was also determined for binding to linear NCs, also by considering that 5% of N molecules remained phosphorylated and bound P_2_ with a high affinity (Figure 7b). These results confirmed that the phosphorylation of RABV N largely enhanced the affinity for P binding and that both high- and intermediate-affinity sites for P_2_ were affected by phosphorylation.

## 4. Discussion

### 4.1. Phosphorylation of Ser389 in RABV N Enhances Binding Affinity for P CTD

We demonstrated here that the phosphorylation of Ser389 in RABV N significantly enhances the binding affinity of P, supporting previous observations [51] and rationalizing our previous structural model of P CTD bound to a polymeric N-RNA complex with phosphorylated N Ser389 [33]. In this model generated by flexible docking and MD simulations using SAXS constraints, the CTD was positioned on the top of the N_CTD_ of the N_i_ protomer and was pinched between the flexible N_CT-ARM_ of the same protomer and of the adjacent protomer (N_i−1_) that molded around the CTD (Figure 8A). Several hydrophobic residues in the N_CT-ARM_ of the N_i_**_−_**_1_ protomer filled a hydrophobic cavity (W-hole) on one face of P CTD, while acidic residues in the N_CT-ARM_ of the N_i_ protomer interacted with the opposite basic face of P CTD [33]. In addition, when N Ser389 was phosphorylated in both N_i_ and N_i_**_−_**_1_ protomers, networks of salt bridges appeared in the MD simulations that suggested additional stability of the complex: N_i_ phosphoserine created new salt bridges with the side chains of P CTD Arg249 and of N_i+1_ Arg418, while N_i__−1_ phosphoserine made a network of salt bridges with the side chains of P CTD Lys211, Lys256, and Arg293 (insets in Figure 8A). In agreement with this model, the partial phosphorylation of N leads us to consider four types of binding sites, with possibly different affinities depending on if one or two phosphoserines are involved in the binding site (Figure 8B): sites of high affinities with two adjacent N protomers phosphorylated on Ser389 (i), sites of intermediate stability with only the N_i_ (ii) or the N_i−1_ (iii) protomer phosphorylated on Ser389, and sites of low affinity with two unphosphorylated N protomers (iv). These different types of binding sites could explain at least part of the complexity observed in the dissociation kinetics.

In RABV-infected cells, phosphorylated N was found in the polymeric N-RNA complex (NC), but not in the soluble, RNA-free N^0^ form associated with P [50]. The absence of the phosphorylation of N^0^ would thus prevent a strong intramolecular interaction between P CTD and N in the N^0^-P complex. Ser389 may not be accessible to casein kinase II in the N^0^-P complex [50], but in the polymeric N-RNA complex, the first half of this N_CT-ARM_ (aa 350–375) becomes stabilized by docking onto the C-terminal domain of the adjacent N_i+1_ subunit [16], possibly exposing the phosphorylation site and allowing kinase access to Ser389. The phosphorylation of N could thus play a role during the replicative cycle by controlling the interactions between P and the NC and therefore affect transcription and replication [45,46].

### 4.2. RABV P Dimerization Moderately Enhances the Affinity for Multimeric N-RNA Complexes

Our results show that the dimeric full-length RABV P protein (P_2_) binds to the purified recombinant circular N_10_-RNA complex and linear nucleocapsid about 15–30-fold better than the monomeric constructs (P_Δ90–133_ and P_CTD_), revealing an important effect of the dimerization for the association of P with the N-RNA complex (difference in binding energy of about 3 RT). The slight difference in affinity between P_Δ90–133_ and P_CTD_ can result from the slow diffusion of P_Δ90–133_ (see above) or indicate the presence of a minor additional binding site in the N-terminal part of the protein (possibly the N-terminal region involved in the formation of the N^0^-P complex [27]). However, a two-fold difference in the dissociation constant contributes only a minor difference of 0.7 RT to the binding energy. The affinity enhancement with respect to monomeric fragments can rather be explained by an avidity effect.

The equilibrium binding isotherms support a model in which the monomeric fragments of P (P_Δ90–133_ and P_CTD_) bind to independent sites, but both SEC and SPR data show that only two CTD can bind to an N_10_-RNA complex [33]. The experimental methods (SPR, SEC) used here to monitor the association of P_2_ with the N-RNA arrays do not allow us to differentiate between dimers that are bound by one or two CTDs, and therefore only measure the total binding of individual P_2_ molecules. In contrast to a simple binding model where one monomeric ligand binds to one monomeric receptor, in our system, different CTD binding sites exist along the N-RNA array, and once one CTD is bound, there are still different sites available for binding the second CTD.

In such a model, if each CTD binds with the same microscopic affinity to any possible binding site of the N-RNA lattice, the enhancement of the apparent affinity for P_2_ can be explained if we consider that both CTDs of each P_2_ can bind to the N_10_-RNA circular complex [55]. If binding of the first CTD occurs with the same affinity (K_a_ = 80 nM) as isolated P_Δ90–133_ (or P_CTD_), the higher apparent affinity originates from the additional stability provided to the N_10_-RNA-P_2_ complex by the intramolecular association of the second CTD ($K_{b}^{i}$ = 0.03) (Figure 5). Although the isotherm could be reproduced using the same parameters for both P_2_ molecules (K_d,app_ = 5.5 ± 1.0 nM), a slightly better fit could be obtained by considering that the second P_2_ molecule binds with a lower affinity than the first (Figure 5). The effective concentration given by the product $K_{a}{.K}_{b}^{i}$ is 2.7 μM for the binding of the first P_2_ and 16.8 μM for the second. The slight reduction in apparent affinity (which we modeled by an increase in K_a_, but could as well be modeled by changing $K_{b}^{i}$ or both constants) could be explained by a reduction in the number of available binding sites once the first P_2_ molecule is bound as described for ligand binding on the lattice [56]. This reading of the data implies that four CTDs and possibly even more (we found that one or two additional P_2_ molecules bind to the same complex with micromolar affinities) bind to a single circular N_10_-RNA complex, as demonstrated by our SEC [33] and SPR data, but in contradiction with the results expected from the experiments with the monomeric fragments.

The magnitude of this affinity enhancement is, however, less than that expected for a bivalent ligand binding to multimeric independent sites. The connection by a flexible linker of two ligands that bind to independent sites has been exploited to increase the effective affinity, in particular of antibodies [57,58] or DNA binding proteins [59]. According to a model developed by Zhou, the effective concentration in such a system can be predicted by calculating the probability density of the end-to-end vector of the flexible linker (using a wormlike chain model) when the ligands are bound to their receptor [60,61]. In the complex formed with the N_10_-RNA circular complex, our structural model based on SAXS data predicted that the CTDs bind with their N-terminus oriented towards the inside of the ring and that the distance between the linker ends is of the order of 2.7 nm [33]. Considering distance and the length of linkers as the sum of the two flexible linkers connecting MD to CTD (aa 133–193) in P_2_, the effective concentration predicted by Zhou’s model is of the order of 1 mM [60,61]. According to this prediction, the intramolecular dissociation constant $K_{b}^{i}$ should be 0.00008 and the first P_2_ molecule should bind with an apparent binding affinity (${K_{a}\cdot K}_{b}^{i}$) of 6.4 pM. These predicted values contrast strikingly with those determined experimentally, probably reflecting steric hindrance and/or conformational constraints in P_2_ and indicating a strong energetic penalty for binding the second CTD. Thus, the dimerization of P only provides a moderate enhancement of apparent affinity for the N-RNA complexes as compared to the theoretical potential.

Why the P protein of MNV members is multimeric remains a matter of debate. It has been shown previously that a mutant of RABV P deleted from the central dimerization region was still active in transcription in a minireplicon assay [62], suggesting that the dimerization domain of RABV P is dispensable for RNA synthesis in vitro, at least for transcription. Similar results have been obtained for VSV [63,64]. In contrast, the RABV P dimerization domain was found to be essential for inducing liquid–liquid phase separation and for the formation of the membrane-less compartments where viral RNA synthesis takes place [35,65], indicating that this domain is essential for viral replication in a cellula.

There are stages in the viral cycle in which the formation of a stable complex between P and NC may be essential for the virus. First, for viral particle assembly, according to the cartwheeling model [42] and the structure of L [32], one P dimer is attached to each polymerase molecule, but a large number of P molecules are bound to the NC in the viral particles [66]. Although the exact role of these extra P molecules is unknown, the tight binding of CTD to phosphorylated NC would allow for recruiting P_2_ to the nascent NC and allow the assembly of a stable ribonucleoprotein complex, preceding its incorporation into a new viral particle. Second, for the initial stage of the viral cycle, when the ribonucleoprotein complex is released into the cytoplasm of infected cells, at that stage, the cellular concentration of soluble P is low or even non-existent. Tight binding of P_2_ to the viral NC because of P dimerization and N phosphorylation could ensure that P_2_ remains bound to the template during the early stages of the replication cycle in the infected cell and may also be used for initiating phase separation.

### 4.3. Interactions Between RABV P and Polymeric N-RNA Complexes

Interactions between multivalent ligands and linear lattices of multiple binding sites are important systems in biology but their binding behaviors can be complex, making a detailed analysis difficult [56]. In this case, we are studying the binding of bivalent RABV P either to a circular N-RNA lattice of 10 N subunits (N_10_-RNA complex), or to a mixture of a linear lattice (NC) containing an heterogeneous (and unknown) number of N subunits. From our previous model of the complex between P CTD and a circular N-RNA complex [33], we know that the binding site for P CTD straddled on two adjacent N protomers, suggesting that a 10 mer circular complex could accommodate up to five CTDs in this binding mode. Our RABV system is further complicated by the phosphorylation of only a fraction of the binding sites on N, which creates high-affinity sites for P CTD and potentially different types of sites of different affinities (Figure 8B). As previously observed for the binding of P_CTD_ [33], both association and dissociation processes of P_2_ and P_Δ90–133_ to circular and linear N-RNA complexes exhibited complex kinetic behaviors when monitored by SPR that could not be fitted with simple models. This complexity may be explained by the “car parking” problem typical of multivalent ligand binding on a lattice of binding sites [56], and by the large dimensions of P due to its flexibility (see above), but also by the existence of binding sites with different affinities resulting from the partial phosphorylation of N (Figure 8B) or by mass transport effects.

The binding association occurs on a similar timescale for all forms of P, suggesting that the differences found in binding affinity are mainly explained by differences in off-rates. For all three proteins, the dissociation kinetics from the highest-affinity binding sites were too slow to be monitored, and k_off_ values could not be determined. In the case of full-length P_2_, we hypothesized that two P dimers were bound to these high-affinity phosphorylated sites of the circular N_10_-RNA complex (complexes I and II in Figure 2A) and dissociated too slowly to be monitored (Figure 3). With CTD binding using two adjacent N protomers, the binding of two P_2_ occupies up to 8 N protomers, mainly protomers in which N Ser389 is phosphorylated. In contrast, the dissociation of one or two additional P_2_ molecules bound to low-affinity non-phosphorylated sites (complex III) gave rise to the observable part of the dissociation kinetics. The probably random distribution of phosphorylated N within circular and linear complexes (average: 70%) generates heterogeneity in the number and position of high-affinity binding sites, which must affect the shape of the dissociation kinetic curve.

The complexity of the binding association and dissociation kinetics may also reflect a mass transport limitation. In a situation where the association rate (k_on_) is of the same order of magnitude as the diffusion rate, a large amount of the dissociated ligand will rebind before diffusing away from the receptor, resulting in an apparent slow dissociation or an apparent absence of dissociation. The presence of long intrinsically disordered regions imposes a considerable viscous drag on the motion of the protein. The translational diffusion coefficient of RABV P_2_ (D value of 4.5 ± 0.1 × 10^−7^ cm^2^ s^−1^) previously measured by dynamic light scattering and analytical ultracentrifugation [24] was lower than expected for a protein of this molecular mass (a theoretical value of 6.3 × 10^−7^ cm^2^ s^−1^ can be calculated for a protein of 68 kDa from a plot of log D versus log MM of globular proteins) [67]. This suggests that the slow diffusion of P_2_ or P_Δ90–133_ molecules and their rebinding to high-affinity sites can slow down the dissociation from the complex formed with N-RNA and could, for example, explain the factor of two observed between the affinity of P_CTD_ and P_Δ90–133_.

Although our results did not permit establishing a complete mechanism describing all of these interactions, they do shed new light on the effects of P dimerization and N phosphorylation on the interactions between P and polymeric N-RNA complexes.

### 4.4. Relevance of RABV P-N Interaction Dynamics for Polymerase Motion

The RNA-dependent RNA polymerase is a processive enzyme [68,69] that must remain attached to the N-RNA template while it progresses along it. As the P protein in all NNVs has been found to be multimeric and to bind both the polymerase and the N-RNA template, it is currently accepted on the basis of work primarily performed with the Sendai virus that the P multimer moves in a cartwheel fashion along the template [42,70]. In this process, the CTD of one P multimer continually makes and breaks contacts with successive N protomers, while the central multimeric domain undergoes a concerted rotational movement [42,70]. Although the exact mechanism has yet to be established, it is likely that the P multimers would cartwheel by being dragged along by the polymerase as the latter moves along its template, rather than being the motor moving the polymerase. P would then act as a processivity factor that keeps the polymerase attached to the template.

Every RABV N protomer binds nine nucleotides [16]; thereby, about 1300 copies of N are required to form an NC, which covers the entire genome (about 12,000 nt) [66]. NNV polymerases synthesize RNA at a rate that varies between one and five nucleotides per second, depending on the virus and the available results (VSV: 3.1–4.3 nt.s^−1^ [71], Sendai virus: 1.7 nt.s^−1^ [72], measles virus: 3 nt.s^−1^ [73]). In line with these data, we assume that it takes 2 to 5 s for the RABV polymerase to assemble 9 nt. In a model in which P cartwheels along the N-RNA template during RNA synthesis, each CTD should then dissociate from its binding site on the N-RNA complex and re-associate on the next binding site roughly every time that the polymerase copies 9 nt. This implies a k_off_ value of the order of 0.2–0.5 s^−1^. The off-rate measured here for the dissociation of P from the dephosphorylated N-RNA template is of this order of magnitude (0.12 s^−1^), suggesting that the dynamics of interaction between P and low-affinity unphosphorylated sites is compatible with a mechanism in which P dimers progress with the L polymerase. In contrast, the off-rate for high-affinity sites in the phosphorylated N-RNA template was too low to be measured over the time of our experiments. We estimated an upper limit of 0.001 s^−1^ for the off-rate of CTD, which is more than two orders of magnitude slower than expected to keep pace with the polymerase.

This paradox raises questions about the mechanism of polymerase motion along the N-RNA template, as well as the function and timing of N phosphorylation. Based on similar arguments, we previously proposed an alternative model to the cartwheeling of P, in which P dimers would attach at regular intervals along the N-RNA template and the polymerase would progress by jumping from one P to the next [33]. Alternatively, P dimers could be tightly bound to the phosphorylated NC, and cartwheeling during RNA synthesis could be triggered and regulated by the dephosphorylation of the template on the passage of the polymerase. Although neither casein kinase nor any cellular phosphatase has been found in the RABV virion [74], different cellular kinases and phosphatases are involved in the regulation of the viral transcription of different NNVs [75,76,77], and this needs to be investigated for RABV. In conclusion, interactions between P and N-RNA polymeric complexes are a potential target for the development of new antiviral compounds, and we hope that our quantitative study of these interactions will pave the way for this approach.

## Figures and Tables

**Figure 1 viruses-16-01735-f001:**
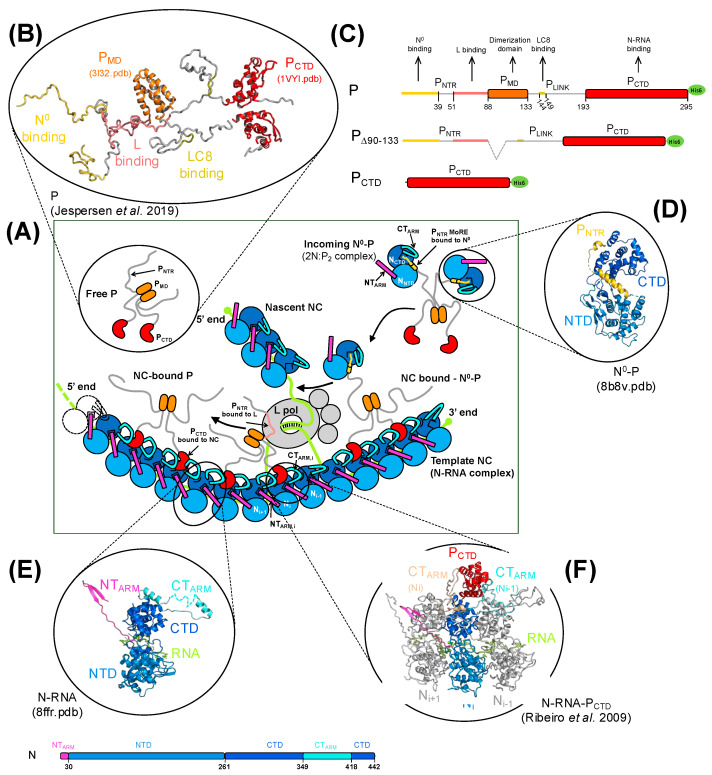
The RNA synthesis ribonucleoprotein complex of RABV. (**A**) A schematic representation of the RABV RNA synthesis ribonucleoprotein complex in replication mode. The nucleoprotein globular domains (N_NTD_ and N_CTD_) are shown in two shades of blue. The N_NT-ARM_ and N_CT-ARM_ are shown in magenta and cyan, respectively. The RNA is shown in green. The L polymerase is shown in gray, while dimeric P is shown in gray with its P_MD_ in orange, and P_CTD_ in red. P is attached to N^0^ through the first part of its P_NTR_ (shown in gold) and to NC through its C-terminal domain (P_CTD_). P is also attached to L through the second part of its P_NTR_ (shown in pink). (**B**) A structural model of RABV P showing its folded domains (P_MD_ in orange and P_CTD_ in red in cartoon representation). The binding regions for N^0^, L, and LC8 in its intrinsically disordered regions are shown as lines. The model is a conformer taken from an ensemble model reproducing SAXS data [20]. (**C**) The architecture of RABV P constructs used in this study showing the long N-terminal intrinsically disordered region (P_NTR_), the multimerization domain (P_MD_, in orange), and the NC-binding domain (P_CTD_, in red) connected by a short flexible linker (P_LINK_). Boxes represent folded domains of known structure and lines represent intrinsically disordered regions. Functional modules are colored and indicated by arrows. (**D**) The structure of the N^0^-P complex. The cartoon representation of the crystal structure of RABV N^0^-P with N in blue and the N-terminal chaperone module of P_NTR_ in gold [27]. (**E**) The architecture of RABV N. The cartoon representation and schematic representation of a protomer of N extracted from the crystal structure of the circular N_11_-RNA complex [16] showing the two globular domains, N_NTD_ and N_CTD_, and the two subdomains, N_NT-ARM_ (in magenta) and N_CT-ARM_ (in cyan), which exchange with adjacent subunits in polymeric N. (**F**) The N-RNA-P_CTD_ complex. The cartoon representation of the model of P CTD bound to the polymeric N-RNA complex showing the N_CT-ARM_ of the adjacent N_i_ and N_i−1_ protomer molding around P CTD [33].

**Figure 2 viruses-16-01735-f002:**
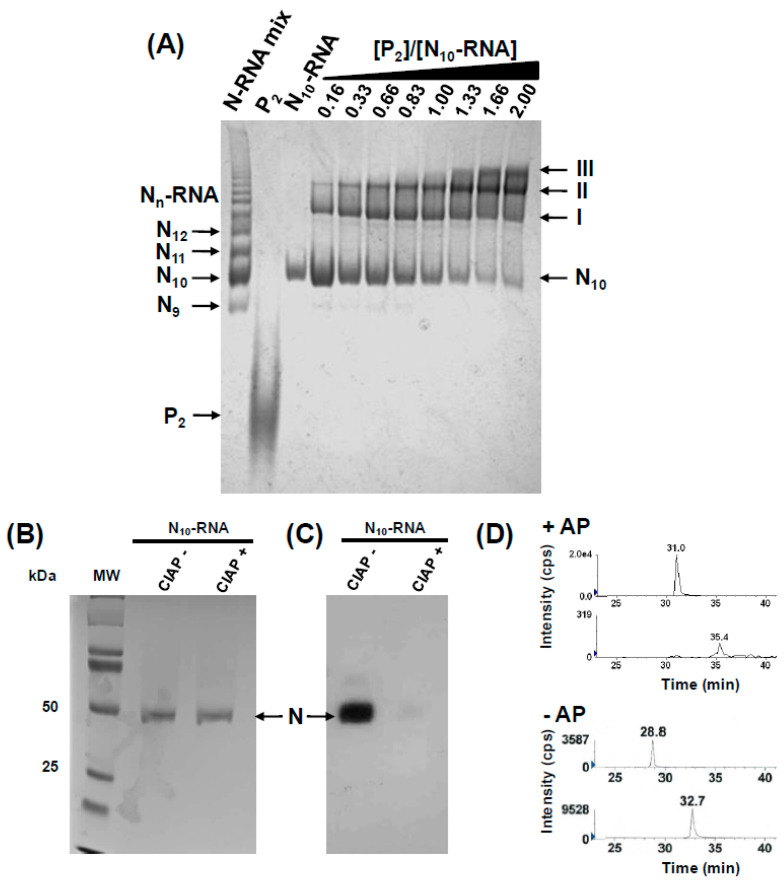
The existence of binding sites of different affinities revealed by the titration of the circular N_10_-RNA complex by full-length P_2_. (**A**) Electrophoresis in native gel. The different circular N_n_-RNA species were separated by native gel electrophoresis (4%). Lane 1 is a mix of different N_n_-RNA rings, where the number of protomers of N in each ring is indicated as a subscript. Lane 2 was loaded with 10 μM of P and lane 3 with 2 μM of purified N_10_-RNA. [P_2_] and [N_10_-RNA] at different ratios were mixed and incubated during 10 min at room temperature and loaded in lanes 4 to 11. In lane 4, [P_dimer_]/[N_10_-RNA], is 0.16, in lane 5 is 0.33, in lane 6 is 0.66, in lane 7 is 0.83, in lane 8 is 1.00, in lane 9 is 1.3, in lane 10 is 1.66, and in lane 11 is 2.00. (**B**) SDS-PAGE of the nucleoprotein before and after phosphatase treatment. The RABV circular N_10_-RNA complex purified from insect cells was treated with calf intestine alkaline phosphatase (CIAP). The reaction mixtures were denatured and applied to 10% SDS-PAGE, followed by Coomassie staining. The first lane (MW) contains a molecular weight marker. (**C**) The Western blot. The proteins separated by SDS-PAGE were blotted onto a nitrocellulose filter. The membrane was incubated with the anti-phosphoserine antibody conjugated to biotin and revealed by a treatment with a streptavidin–horseradish peroxidase conjugate. (**D**) Mass spectrometry fingerprints. The chromatographic profiles show the intensity of the signal for the unphosphorylated (upper trace) and phosphorylated forms of the peptide (lower trace) after treatment with the alkaline phosphatase (+AP) or without treatment (−AP).

**Figure 3 viruses-16-01735-f003:**
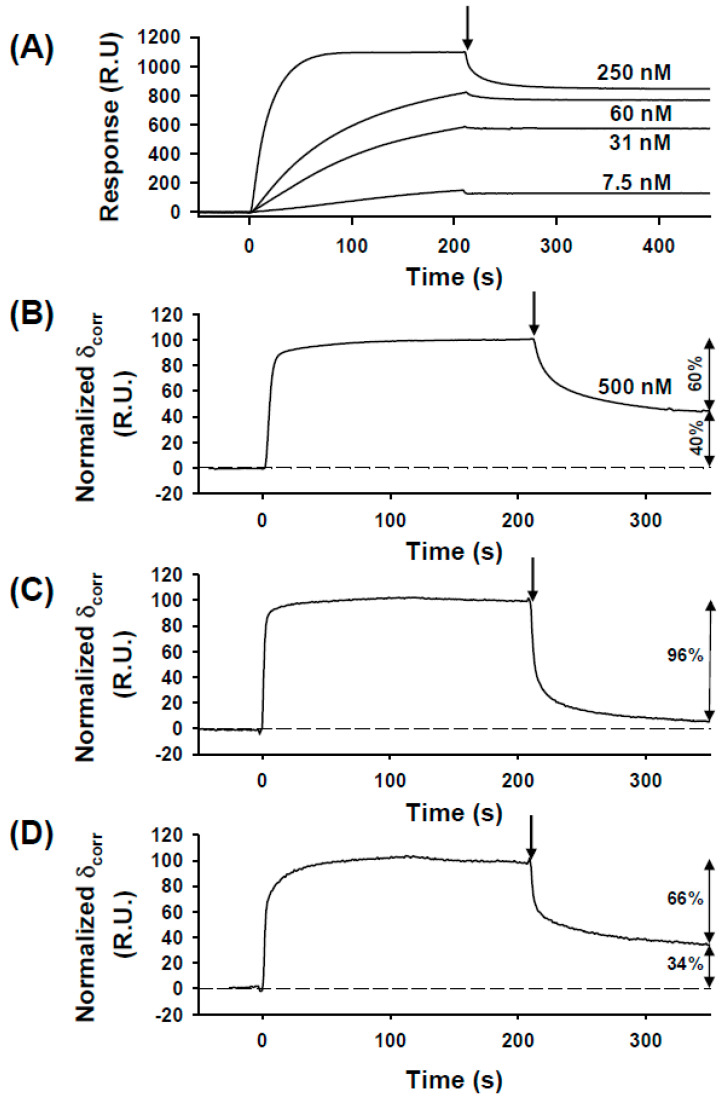
Interactions between the full-length P dimer (P_2_) and circular N_10_-RNA complex by surface plasmon resonance. In each graph, the arrow indicates the initiation of dissociation by flowing the buffer over the surface. The concentration of P_2_ is shown above the lines and the double arrows on the right indicate the amplitude of dissociation processes whose kinetics may or may not be measurable. (**A**) The sensorgram (response in resonance units (R.U.) versus time) of the binding of RABV P_2_ to the immobilized N_10_-RNA complex. The N_10_-RNA complex was immobilized on a CM5 sensor chip as described in the Section 2. The full-length P dimer (P_2_) was injected over the immobilized N_10_-RNA complex at different concentrations in the 20 mM Tris-HCl buffer at pH 7.5 containing 150 mM NaCl, 1 mM DTT, and 0.005% (*v*/*v*) Tween20 at 20 °C and at a flow rate of 20 µL·min^−1^. The specific binding signals were obtained after the subtraction of a background signal measured in a reference channel with no bound N_10_-RNA complex. (**B**) The sensorgram of binding full-length P_2_ to the N_10_-RNA complex purified from insect cells. Full-length P_2_ was injected at a concentration of 500 nM. Response was normalized at a signal of 100 R.U. at equilibrium. (**C**) The sensorgram of binding full-length P_2_ to dephosphorylated N_10_-RNA after treatment with alkaline phosphatase (CIAP). (**D**) The sensorgram of binding the full-length P dimer to re-phosphorylated N_10_-RNA after treatment with casein kinase II (CKII).

**Figure 4 viruses-16-01735-f004:**
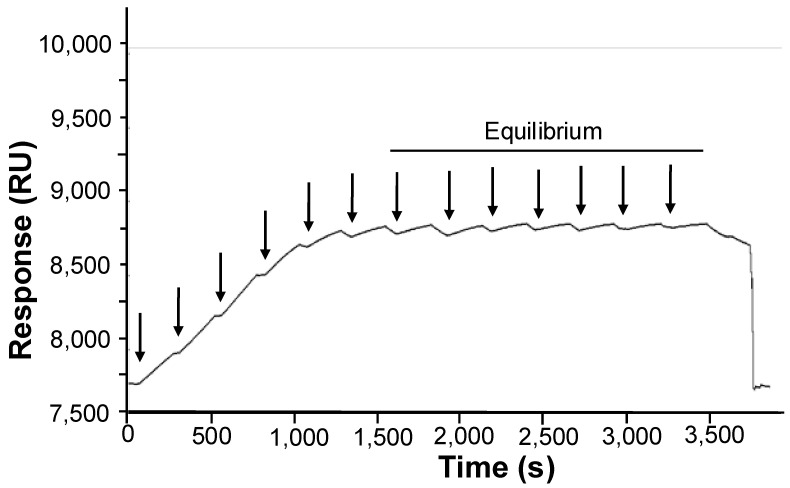
The equilibrium SPR analysis using multiple injections. The N_10_-RNA complex was immobilized on a CM5 sensor chip as described in the Section 2. Full-length P_2_ was injected at a concentration of 50 nM in the 20 mM Tris-HCl buffer at pH 7.5 containing 150 mM NaCl, 1 mM DTT, and 0.005% (*v*/*v*) Tween20 at 20 °C and at a flow rate of 20 µL·min^−1^. Successive injections of the ligand are shown by arrows. Equilibrium was reached after approximately 25 min and the signal remained unchanged during subsequent injections.

**Figure 5 viruses-16-01735-f005:**
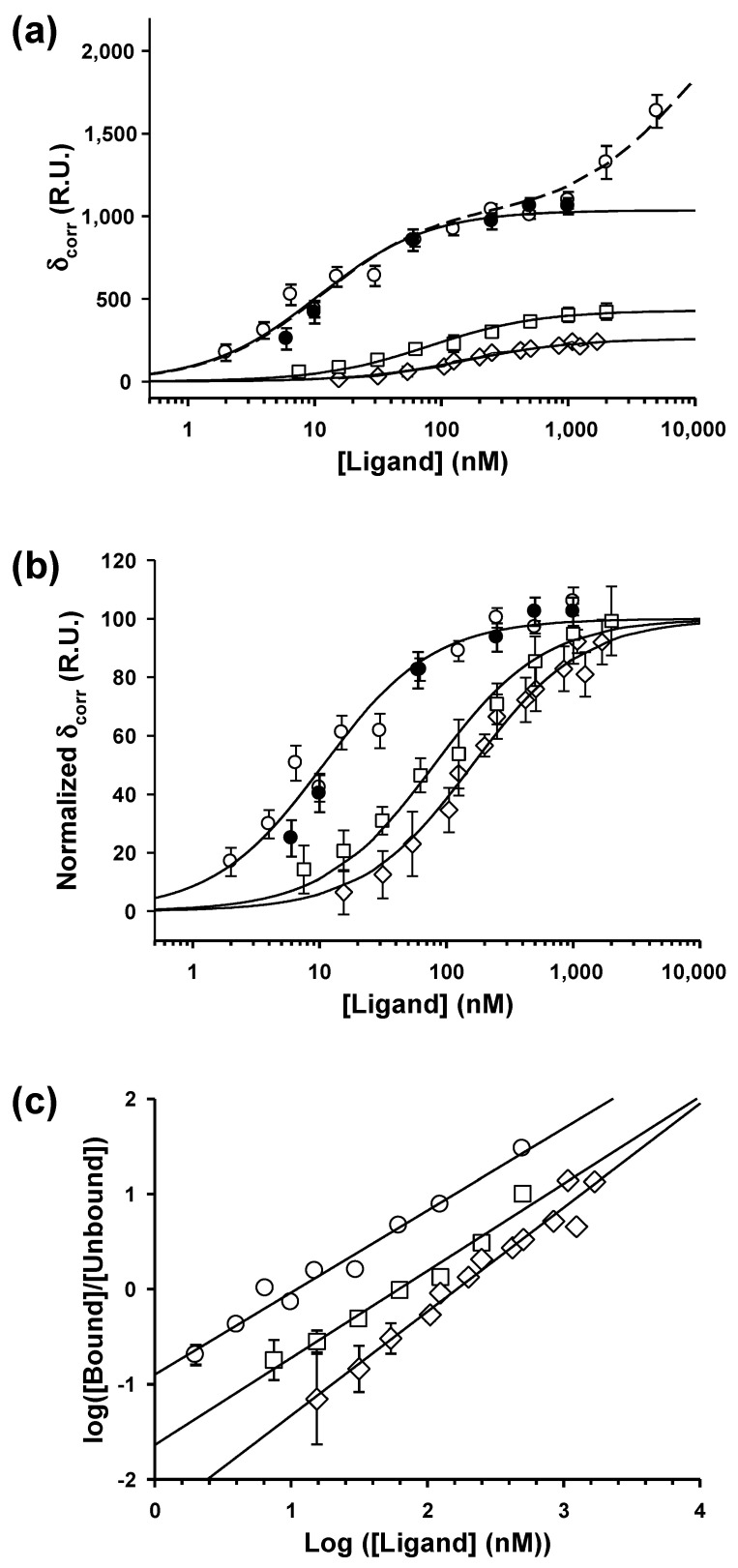
Equilibrium binding interactions with circular N_10_-RNA complexes. (**a**) Binding isotherms of full-length P_2_ (circles), P_CTD_ (diamonds), and P_Δ90–133_ (squares). The SPR signal at equilibrium is shown as a function of ligand (L) concentration. Open symbols are in the presence of 150 mM NaCl and closed symbols are in the presence of 500 mM NaCl. The dotted lines show the fit for independent-site binding. The solid line shows the fit to binding model 3 using two binding terms and the dashed lines show the fit using three binding terms, assuming that two additional P2 bind to low-affinity sites. (**b**) Normalized binding isotherms. The data of panel (**a**) below 1 μM were normalized to a saturation value of 100 R.U. The symbols are the same as in panel a. The dotted lines show the fits to an independent site, and the solid line shows the fit to model 3 for the first two P_2_ molecules. (**c**) Hill plots. The symbols are the same as in panel (**a**). The dotted lines show the fits to an independent site, and the solid line shows the fit to model 3 for the first two P_2_ molecules.

**Figure 6 viruses-16-01735-f006:**
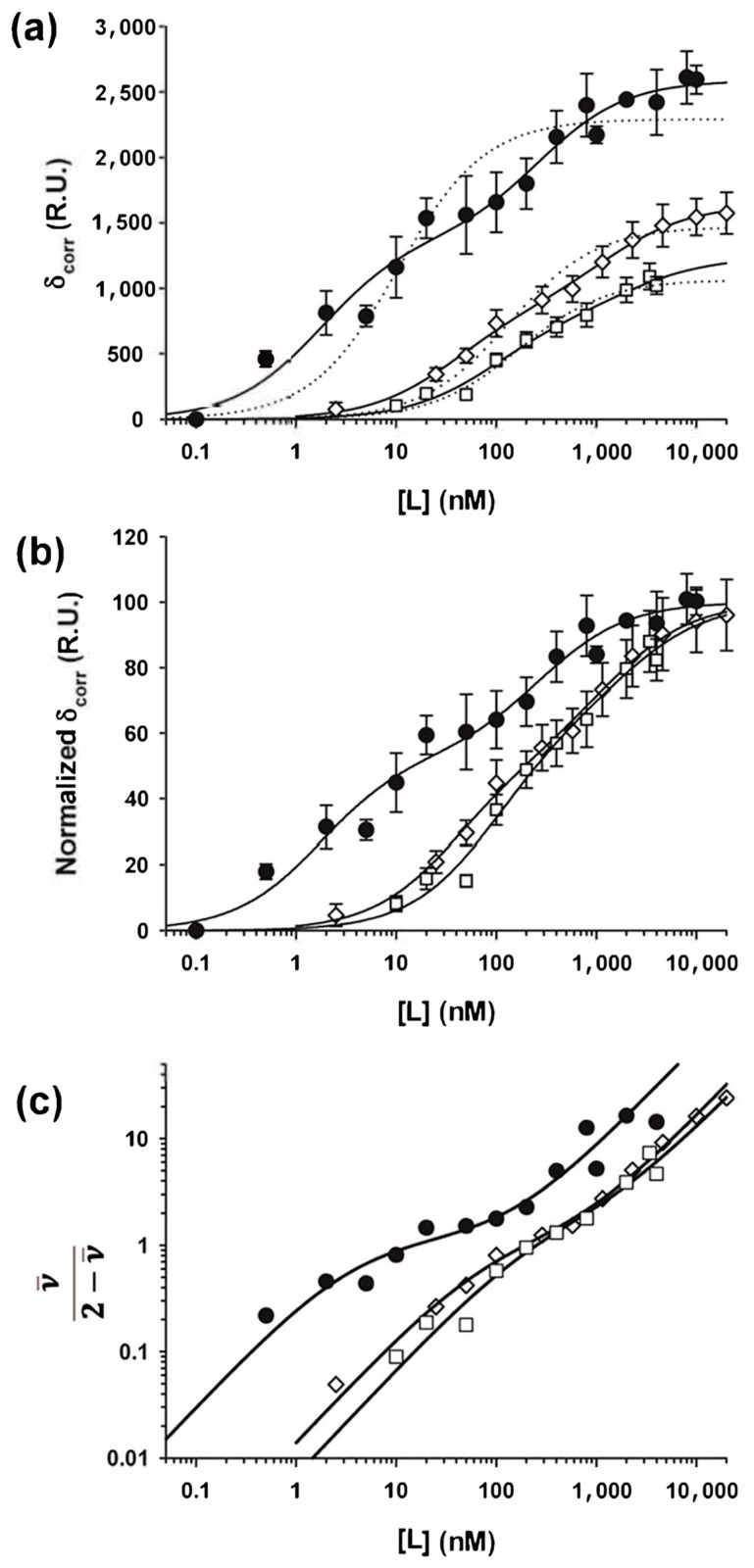
Equilibrium binding interactions with linear NCs. (**a**) Binding isotherms of full-length P_2_ (circles), P_CTD_ (diamonds), and P_Δ91–131_ (squares). The SPR signal at equilibrium is shown as a function of ligand (L) concentration. The dotted lines show the fit for binding two ligands with two independent-site bindings using the same dissociation constant for both. The solid lines show the fit for binding two ligands with two independent-site bindings using different dissociation constants. (**b**) Normalized binding isotherms. The data of panel (**b**) were normalized to a saturation value of 100 R.U. The symbols are the same as in panel (**a**). The lines show the fits to an independent site using different binding constants. (**c**) Hill plots. The symbols are the same as in panel (**a**). The lines show the fits to an independent site.

**Figure 7 viruses-16-01735-f007:**
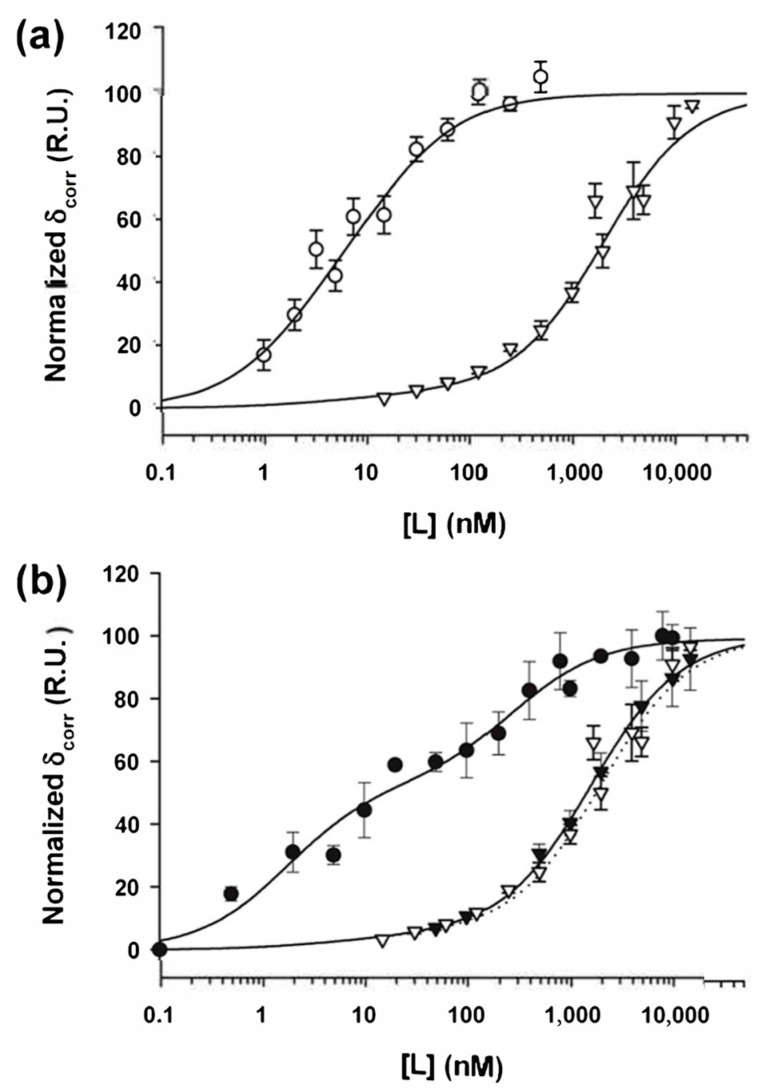
Equilibrium binding interactions with the dephosphorylated circular N_10_-RNA complex and linear NCs. The N10-RNA complexes or the NC were first bound to the SPR chip and then were treated with CIAP. The SPR signal at equilibrium is shown as a function of ligand (L) concentration. (**a**) Normalized binding isotherms of full-length P_2_ to circular N_10_-RNA complexes. The empty circles show the data obtained with the phosphorylated N_10_-RNA complex taken from Figure 5b, together with their fit (shown as a line). The empty triangles show the data obtained with the dephosphorylated N_10_-RNA complex and the line shows the fit including 5% of P_2_ bound to the remaining high-affinity phosphorylated sites. (**b**) Normalized binding isotherms of full-length P_2_ to linear NC. The filled circles show the data obtained with the phosphorylated linear NC taken from Figure 5b, together with their fit (shown as a line). The filled triangles show the data obtained with dephosphorylated linear NC and the line shows the fit including 5% of P_2_ bound to the remaining high-affinity phosphorylated sites. The empty triangles and the dotted lines show the data and their fit obtained with dephosphorylated N_10_-RNA complexes taken from panel (**a**) as comparison.

**Figure 8 viruses-16-01735-f008:**
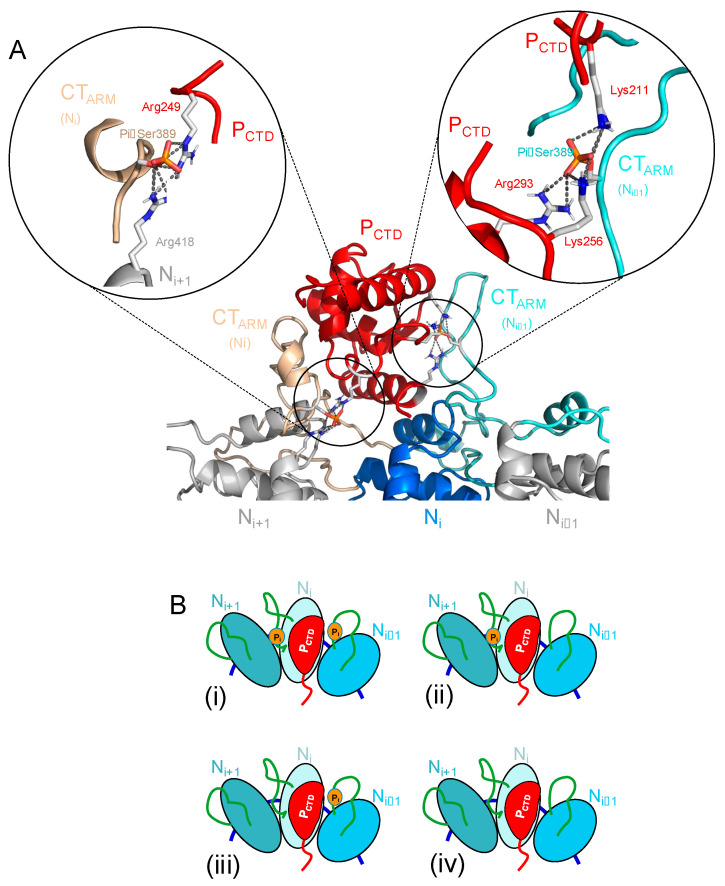
A structural model of the phosphorylated sites in RABV N-RNA. (**A**) A structural model of CTD in complex with the Ser389-phosphorylated N-RNA complex. The model was generated by computational modeling using SAXS data [33]. P CTD and the core of three adjacent N protomers are shown in the cartoon representation. Ser389 is phosphorylated in the N_i_ and N_i−1_ protomers, and the residues that were found to form salt bridges (gray dotted lines) with the phosphate group during the MD simulations are shown in the insets and are labeled. (**B**) Schematic representations of one CTD bound to the N_i_ protomer in the N-RNA complex that show the four types of possible binding sites depending on the phosphorylation of Ser389 in the N_i_ and N_i−1_ protomers. (**i**) Ser389 in the N_CT-ARM_ of the N_i_ and N_i-1_ protomers are phosphorylated. (**ii**) Only Ser389 the N_CT-ARM_ of the N_i_ protomer is phosphorylated. (**iii**) Ser389 in the N_CT-ARM_ of the N_i-1_ protomer is phosphorylated. (**iv**) Ser389 in neither the N_i_ or N_i-1_ protomer is phosphorylated.

## Data Availability

Data available upon request.

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
