# Peer review of "Dimerization of Rabies Virus Phosphoprotein and Phosphorylation of Its Nucleoprotein Enhance Their Binding Affinity"

_viruses, 2024, doi:10.3390/v16111735_

Round 1

Reviewer 1 Report

Comments and Suggestions for Authors

Dear Editor, dear Authors,

The phosphoprotein (P) of mononegaviruses plays a pivotal role in the viral cycle, mainly via its function as the polymerase cofactor. P mediates multiple and dynamic interactions with several cellular and viral proteins.  Understanding the characteristics and dynamics of these interactions is essential to decipher the molecular mechanisms used by these viruses to transcribe and replicate their genome. Notably, the C-terminal domain of P (PCTD) binds the encapsidated genome, or nucleocapsid (NC), and thus tether the viral polymerase on its template. 

In this work, the authors studied the interaction between the PCTD and NC of rabies virus, a member of the Mononegavirales order, and a significant concern for human health. They purified the protein complexes and mainly used SPR to characterize this interaction. 

The conclusions mostly validate previous results but with more precision. 

The article isn’t easy to follow. The discussion section is very long and sometimes deals with questions unrelated to the experiments of the project (such as the 16-line-long paragraph on a hypothesis of why the N is phosphorylated on the NC and not in the N0-P complex). 

I do not have major concerns about the experiments or the results. 

Here are a few points that, if addressed, may improve this interesting work. 

-        It would be useful to include an introduction schematic of the N and P proteins to show their domain organization and interaction sites. 

-        Regarding lines 251-252, it would be nice to add a gel showing the re-phosphorylation of the N protein. 

-        Lines 491-492, why do you consider that two additional P2 are binding the ring with lower affinity? Why two and not one P2? This would make 4 P2 per ring but since we can’t see a fourth band on the native gel, it would mean that a ring with 3 P2 does not exist. Please clarify. 

-        It is claimed that the non-circular nucleocapsid is “helical”. Why? We know the conformation of the nucleocapsid of rhabdoviruses can adopt different conformations based on the salt concentration and pH. Is it known that RABV NC is helical under the conditions used? It would be great to see the NC by negative stain EM. The helical conformation is an important point to clarify since the curvature of the NC may affect the binding of P. Else, the “helical” NC should be renamed “linear” NC as opposed to the circular NC: the rings. 

-        Based on the model presented in Figure 7, is there enough space for one PCTD per N? If not, in theory, based on the structure model, how many PCTD-binding sites are there per N10 ring? How does it fit with the data?

-        It would be useful to include a conclusion schematic representing the results of the study.

Also, 

-        Lines 250-251, Figure “1b” and “1c” are inverted. 

-        Text is missing in Figure 2 (may be an issue with my version of the manuscript).

-        Line 767, replace “cellular” with “viral”.

Regards.

Author Response

We thank the reviewers for their careful reading of our manuscript and for their corrections and suggestions for improvement. Here are the detailed answers to their requests.

Answer to referee #1’s comments

The article isn’t easy to follow. The discussion section is very long and sometimes deals with questions unrelated to the experiments of the project (such as the 16-line-long paragraph on a hypothesis of why the N is phosphorylated on the NC and not in the N0-P complex). 

We've reorganized the discussion text and removed certain sections, in the hope of making it easier to read.

-        It would be useful to include an introduction schematic of the N and P proteins to show their domain organization and interaction sites. 

We have added a Figure 1 to describe the structural architecture of the N and P and their functional domains and to describes the two major interactions between N and P

-        Regarding lines 251-252, it would be nice to add a gel showing the re-phosphorylation of the N protein. 

We agree with the referee but this experiment has not been performed and we are currently unable to prepare the material for repeating this experiment. The evidence we have of re-phosphorylation is the restoration of dissociation kinetics shown in Figure 2C.

-        Lines 491-492, why do you consider that two additional P2 are binding the ring with lower affinity? Why two and not one P2? This would make 4 P2 per ring but since we can’t see a fourth band on the native gel, it would mean that a ring with 3 P2 does not exist. Please clarify. 

We agree with the referee that the native gel (Figure 1) indicates the presence of at least one additional species but is not sufficiently resolved to claim the presence of two species. A second evidence for the presence of additional binding comes from the binding isotherm shown in Figure 4A, which demonstrates additional binding at ligand concentrations above 1 mM, but again we agree that because the plateau could not be reached (for solubility problems).

The only evidence that led us to choose the binding of two P2 molecules rather than one to low-affinity sites is the isothermal binding curve in Fig. 5A, which appears to continue to rise above the expected signal for a single molecule (the dashed line was calculated for the binding of 2 additional P2 molecules). However, we agree that these data are not really sufficient to validate this choice. We have retained the model with 2 additional molecules, but modulated our statement accordingly (paragraph lines 536-546).  

-        It is claimed that the non-circular nucleocapsid is “helical”. Why? We know the conformation of the nucleocapsid of rhabdoviruses can adopt different conformations based on the salt concentration and pH. Is it known that RABV NC is helical under the conditions used? It would be great to see the NC by negative stain EM. The helical conformation is an important point to clarify since the curvature of the NC may affect the binding of P. Else, the “helical” NC should be renamed “linear” NC as opposed to the circular NC: the rings. 

We thank the referrer for pointing out this inaccuracy. We have made the necessary corrections by replacing the term “helical” with “linear” throughout the text.

-        Based on the model presented in Figure 7, is there enough space for one PCTD per N? If not, in theory, based on the structure model, how many PCTD-binding sites are there per N10 ring? How does it fit with the data?

In the model that we generated by flexible docking and published in 2009 (Ribeiro et al. J. Mol. Biol.), the C-terminal domain of P (PCTD) sits on the top of one N subunit (Ni) and is flanked by the “arms” from the same subunit (Ni) and from an adjacent one (Ni-1) (the arms are flexible in the absence of PCTD). In this model, there was thus clearly not enough space for one PCTD per N, and only 5 binding sites could be theoretically considered. Our experimental SAXS data obtained for binding of PCTD to circular N-RNA complexes could be reproduced by considering the binding of only two PCTD and the docking of more than two crystal structures of RABV PCTD on the ring led to steric clashes between the N-terminal parts of bound PCTDs. However, the N-terminal part of PCTD crystal structure (residues 186-194) is stabilized by crystal contacts and unfolds quickly in MD simulation.

We have added some description of this model, in the introduction and discussion in the hope that it will help understanding the role N phosphorylation.

-        It would be useful to include a conclusion schematic representing the results of the study.

We hope that the addition of Figure 1 and the modifications of Figure 7 (now Figure 8) will provide together with our modifications of the discussion a clearer pictures of our results

Also, 

-        Lines 250-251, Figure “1b” and “1c” are inverted. 

Corrections have been made.

-        Text is missing in Figure 2 (may be an issue with my version of the manuscript).

We're sorry, but a problem seems to have gone unnoticed when we generated the pdf file in which figure 2 is incomplete (lacking much of the text), although complete in the docx version.  The figure (now Figure 3) is now correct.

-        Line 767, replace “cellular” with “viral”.

 Correction has been made.

Reviewer 2 Report

Comments and Suggestions for Authors

The manuscript devoted to the study of the mechanism of action of the RNA complex. The research topic is relevant. The rabies virus is still fatal in 100% of cases. The article contains a large and interesting introduction. However, its perception is hampered by the lack of illustrations. In particular, lines 52-74 and lines 99-100 should be illustrated.

Figure 2 is uninformative. There are no labels for the axes, which makes it difficult to understand the data presented. When describing the mechanism of protein interaction, it would be helpful to see pictures that show key amino acids and the types of intermolecular interactions involved.

I would recommend including the results of molecular modeling of the interaction between two proteins in your manuscript.

And most importantly, how can the results of this work help researchers in their fight against these viruses? Please provide these arguments in your conclusion.

Author Response

We thank the reviewers for their careful reading of our manuscript and for their corrections and suggestions for improvement. Here are the detailed answers to their requests.

Answer to referee #2’s comments

The article contains a large and interesting introduction. However, its perception is hampered by the lack of illustrations. In particular, lines 52-74 and lines 99-100 should be illustrated.

We have added a Figure 1 to describe the structural architecture of the N and P and their functional domains and to describes the two major interactions between N and P

Figure 2 is uninformative. There are no labels for the axes, which makes it difficult to understand the data presented. When describing the mechanism of protein interaction, it would be helpful to see pictures that show key amino acids and the types of intermolecular interactions involved.

We're sorry, but a problem seems to have gone unnoticed when we generated the pdf file in which figure 2 is incomplete (lacking much of the text), although complete in the docx version. The figure (now Figure 3) is now correct.

I would recommend including the results of molecular modeling of the interaction between two proteins in your manuscript.

The molecular modeling of the interactions between RABV PCTD and N-RNA has already been published in the article by Ribeiro et al. in 2009. We have added a illustrations in Figure 1 to provide some details on the interaction.

And most importantly, how can the results of this work help researchers in their fight against these viruses? Please provide these arguments in your conclusion.

We have added a sentence to this effect at the end of the document.

Reviewer 3 Report

Comments and Suggestions for Authors

Manuscript of Ribeiro et al addresses an important aspect of functioning of the RNA synthesizing machine of rabies virus.

Using surface plasmon resonance and mass spectrometry mainly authors demonstrates essential feature of phosphoprotein and nucleoprotein.

Role of phosphoprotein multimerization for interaction with nucleoprotein in complex with RNA was reported before, but without details.

Authors demonstrates dimerization of phosphoprotein arise affinity to RNA compare to monomer form and measure increasing of affinity of

N-protein to RNA in case of Ser389 phosphorylation. Paper has enough experimental material and good results interpretation and can be interesting

 for readers of Viruses. I recommend publishing it in present sate.

Author Response

We thank the reviewers for their careful reading of our manuscript and for their corrections and suggestions for improvement. Here are the detailed answers to their requests.